# C4d Deposition after Allogeneic Renal Transplantation in Rats Is Involved in Initial Apoptotic Cell Clearance

**DOI:** 10.3390/cells10123499

**Published:** 2021-12-10

**Authors:** Stefan Reuter, Dominik Kentrup, Alexander Grabner, Gabriele Köhler, Konrad Buscher, Bayram Edemir

**Affiliations:** 1Department of Internal Medicine D, Experimental Nephrology, University Clinics Münster, 48143 Münster, Germany; Stefan.Reuter@ukmuenster.de (S.R.); dominik.kentrup@gmx.de (D.K.); alexander.grabner@duke.edu (A.G.); konrad.buscher@ukmuenster.de (K.B.); 2Department of Medicine, Division of Nephrology, The University of Alabama at Birmingham, Birmingham, AL 35233, USA; 3Department of Medicine, Division of Nephrology, Duke University School of Medicine, Durham, NC 27710, USA; 4Gerhard Domagk Institute of Pathology, University Clinics Münster, 48143 Münster, Germany; gabriele.koehler@uni-muenster.de; 5Department of Medicine, Hematology and Oncology, Martin Luther University Halle-Wittenberg, 06120 Halle (Saale), Germany

**Keywords:** complement, C4d, apoptosis, macrophages

## Abstract

In the context of transplantation, complement activation is associated with poor prognosis and outcome. While complement activation in antibody-mediated rejection is well-known, less is known about complement activation in acute T cell-mediated rejection (TCMR). There is increasing evidence that complement contributes to the clearance of apoptotic debris and tissue repair. In this regard, we have analysed published human kidney biopsy transcriptome data clearly showing upregulated expression of complement factors in TCMR. To clarify whether and how the complement system is activated early during acute TCMR, experimental syngeneic and allogeneic renal transplantations were performed. Using an allogeneic rat renal transplant model, we also observed upregulation of complement factors in TCMR in contrast to healthy kidneys and isograft controls. While staining for C4d was positive, staining with a C3d antibody showed no C3d deposition. FACS analysis of blood showed the absence of alloantibodies that could have explained the C4d deposition. Gene expression pathway analysis showed upregulation of pro-apoptotic factors in TCMR, and apoptotic endothelial cells were detected by ultrastructural analysis. Monocytes/macrophages were found to bind to and phagocytise these apoptotic cells. Therefore, we conclude that early C4d deposition in TCMR may be relevant to the clearance of apoptotic cells.

## 1. Introduction

Given the importance of renal transplantation (TX) and the limitation of available donor kidneys, a detailed analysis of factors that shorten graft survival is important. Although significant advances have been made in transplant medicine, (acute) allograft rejection is still a serious problem after TX, threatening graft and patient survival [1]. Allograft rejection is a sterile inflammation of the transplant kidney triggered by under-immunosuppression and mediated by antibodies (ABMR) and/or T cells (TCMR) [2]. The diagnosis is usually based on typical findings in histology (TCMR: mainly tubulitis and arteriitis) or histology and presence of donor-specific antibodies (ABMR: mainly glomerulitis, peritubular capillaritis, thrombotic microangiopathy) [2].

Previous research focused mainly on the T cell-mediated immune response. More recently, the complement system has gained increasing attention because its activation is frequently involved in various deleterious conditions before, during, and after transplantation (comprehensively reviewed in [3]). In renal transplantation (surgery), ischemia and reperfusion of the graft are the most important triggers of complement activation, and the phase of reperfusion is especially critical for the further course. After transplantation, complement is involved in immunologic (mediated by innate immunity) and inflammatory processes that further damage the graft. In addition, complement activation is involved in many different renal diseases, such as complement component *C3* (C3)-glomerulonephritis, IgA nephropathy, lupus nephritis, different thrombotic microangiopathies, and even diabetic kidney disease, which can recur after transplantation, demonstrating the unique susceptibility of the kidney to pathological complement effects [4,5]. Therefore, it was natural to investigate whether complement-targeted strategies play a role in the outcome of renal transplantation, in terms of optimisation of graft quality, as well as in the treatment of ABMR, induction of accommodation, and modulation of the adaptive immune response.

A role of complement as a diagnostic marker for the acute ABMR of renal grafts, described first by Halloran et al. [6], has been elucidated and established by immunohistochemical staining of complement *C4d* deposits by Feucht et al. [7,8]. Recently, studies showed *C4d* deposition also during acute TCMR of transplanted lungs indicating a role of *C4d* as a marker for TCMR [9]. The complement cleavage product C4d is referred to as an immunohistochemical marker for humoral-mediated allo-response [10], while other humoral markers, such as IgG, IgM, and *C3*, were not found to be useful [11]. Consistent with this, a role for *C2* expression and complement activation through the binding of natural alloantibodies has been demonstrated after heart and lung transplantation in the context of ischemia-reperfusion injury (IRI) and rejection [12,13]. In addition, IRI triggers activation of the complement system, promoting inflammation that leads to tubule apoptosis and aggravates renal IRI. Its inhibition or knockdown was shown to decrease the immunologic injury following activation of the complement system [14,15,16]). In line with the literature, we detected increased gene expression of complement factors in human kidney grafts undergoing TCMR [17,18,19]. This led us to two important questions. First, is the increased gene expression associated with activation of the complement system, and second, what is the trigger for its activation?

## 2. Materials and Methods

### 2.1. Animal Model

Male Lewis–Brown-Norway (LBN) and Lewis (LEW) rats (200–270 g body weight (BW), Charles River, Sulzfeld, Germany) with free access to standard rat chow (Altromin, Lage, Germany) and tap water were used (*n* = 4–5/group). Transplantation was performed simultaneously by two investigators, as previously published [20,21]. A detailed (video-based) protocol of a comparable uni-nephrectomised transplant model was published by us in *JoVE* [22]. Kidneys from age- and weight-matched LBN rats were transplanted into LEW rats without immunosuppression to avoid drug-associated effects on kidneys and complement activation. In short, the abdomen was opened by a ventral midline incision, carefully exposing the right kidney and its vessels. Nephrectomy was performed after decapsulation, and the left kidney and its vessels were exposed by moving the intestine to the right side. After nephrectomy of the left kidney, the left donor kidney, including the ureter, the renal artery, a piece of the aorta, and the renal vein, was transferred to the recipient. Transplantations were performed immediately after bilateral nephrectomy of the recipient to mimic the situation in human patients. Surgical experiments were approved by a governmental committee on animal welfare and were performed in accordance with national animal protection guidelines (permit No. 8.87-50.10.36.08.098). While the total operation time of the recipient did not exceed 90 min, the ischemia time of the graft was always shorter than 30 min. Grafts were recovered on postoperative day 1, 2, and 4 (POD4) after transplantation. Because graft necrosis starts only a few days after POD4 due to ongoing rejection in this allogeneic transplant (aTX) TCMR model without immunosuppression, and because it has been shown that one week after aTX, rejection is already more mixed than TCMR, we decided to investigate the changes until POD4 [21,23]. Syngeneically transplanted rats (sTX) without rejection and healthy LBN rats served as controls.

### 2.2. General Functional Data

Serum parameters, total kidney function, and body weight were recorded before transplantation and on POD4. Twenty-four hours before surgery/kidney recovery, animals were housed in metabolic cages. Urine and blood samples were analysed for protein (Bradford Blue, Bio-Rad Laboratories, München, Germany). Electrolytes were analysed by flame photometry (Instrumentation Laboratory 943, Kirchheim, Germany), and creatinine was analysed on a Roche Diagnostics analyser (Modular P, Roche Diagnostics, Mannheim, Germany).

### 2.3. Histology

Portions of kidneys were snap-frozen and fixed in 4% formaldehyde in PBS. Histological changes were examined by light microscopy in paraffin-embedded tissue with periodic acid–Schiff (PAS) staining.

### 2.4. Transmission Electron Microscopy

Transmission electron microscopy experiments were performed on POD4. Kidney slices were washed twice with PBS and fixed in 2.5% cacodylate-buffered glutaraldehyde (pH 7.35). Following postfixation in phosphate-buffered 1.3% osmium tetroxide, tissues were rinsed in buffer, dehydrated in graded alcohol, and processed into polymerised blocks of epoxy resin. Sections for initial light microscopy to determine tissue quality and architecture and suitability for ultrastructure analyses were cut at 0.5 µm and stained with toluidine blue. For electron microscopy, thin sections were cut with a diamond knife, mounted on copper grids, stained with alkaline lead citrate and 8% uranyl acetate, and examined using a Philips EM 208 electron microscope (Philips, Eindhoven, The Netherlands) operating at 80 kV. Photography was performed with a Morada digital camera (Olympus SIS, Münster, Germany).

### 2.5. Microarray Analysis

Microarray analysis was performed as described before [17]. Briefly, control and transplanted kidney tissue were used for total RNA isolation using Qiagen RNAeasy total RNA isolation kits (Qiagen, Hilden, Germany). The used tissue covered all segments of the kidneys, including cortex, medulla, and papilla. The data discussed in this publication have been deposited in NCBI’s Gene Expression Omnibus [24] and are accessible through GEO series accession number GSE6497. Identification of significantly different expressed genes between the allogeneic and syngeneic groups was performed by class comparison using BRB-ArrayTools, developed by Dr. Richard Simon and Amy Peng [25]. The type of univariate test used was a two-sample *t*-test. Exact multivariate permutations tests were computer-based, performed on 126 available permutations. Nominal significance level of each univariate test was set to 0.001. Confidence level of false discovery rate assessment used was 90%, and the maximum numbers of false-positive genes allowed were set to 10. To illustrate the signalling pathways with affected genes in the AR group, the data were analysed and visualised with Ingenuity Pathways Analysis (IPA; Ingenuity Systems, Redwood City, CA, USA).

### 2.6. Real-Time PCR

Expression profiles of selected marker genes were validated by real-time PCR, which was performed using SYBR Green PCR Master Mix or TaqMan Universal PCR Master Mix on an ABI PRISM 7900 Sequence Detection System. Specific primer pairs were used (SDC-1). All instruments and reagents were purchased from Applied Biosystems (Darmstadt, Germany). Relative gene expression values were evaluated with the 2^−^^ΔΔ^^Ct^ method using GAPDH as housekeeping gene [26].

### 2.7. Human Biopsy Transcriptome Analysis

The data set GSE36059 was used (Affymetrix Human Genome U133 Plus 2.0 Array). It includes indication biopsies with *n* = 8 nephrectomies, *n* = 281 non-rejection, *n* = 35 TCMR, *n* = 65 ABMR, and *n* = 22 mixed-rejection [27]. ABMR and mixed-rejection samples were used as positive controls. Nephrectomy samples and non-rejection samples served as negative controls. There was no selection on immunosuppression and time point after transplantation. The complement cascade and apoptosis pathway were extracted from the KEGG database. Pathway activity was calculated by the mean of the sum of all genes in the pathway, expressed as a robust multiarray average (log2 transformed, quantile normalised).

### 2.8. C4d and C3d Deposition

Renal C4d deposition was investigated using a specific antibody for immunofluorescence [28], kindly provided by Dr. W. M. Baldwin III, and an antibody directed against C4d from Hycult Biotech (HP8034; Uden, The Netherlands). C3d deposition was analysed using an anti-C3d antibody (AF2655, R&D Systems, Abingdon, UK). Deposited IgM was detected using an FITC-labelled anti-IgM antibody (# 11-0990-82, Invitrogen, Darmstadt, Germany). Cryosections about 5 µm thick were made from the grafts fixed in melting 2-methylbutane. Cryosections were incubated for 1 h at room temperature in PBS containing 10% normal goat serum (pH 7.4) and then for 90 min at room temperature with affinity-purified primary antibodies at a dilution of 1/200 (kidney sections). After washing in PBS, the sections were incubated for 45 min at room temperature with PBS-diluted secondary antibodies (Alexa Fluor 488 goat anti-rabbit-Ig, 1/1000; Invitrogen, Karlsruhe, Germany). Sections were rinsed with PBS, coverslipped with Crystal Mount (GeneTex, San Antonio, TX, USA), and evaluated by epifluorescence microscopy (Observer Z1 with apotome; Zeiss, Göttingen, Germany).

### 2.9. FACS Analysis

The presence of alloantibody was measured by flow cytometry. Peripheral blood lymphocytes (PBL) from LBN rats were either incubated (60 min) with heat-inactivated (30 min at 55 °C) plasma from recipients or were left untreated. The cells were washed with phosphate-buffered saline. Untreated cells were incubated with an FITC-labelled anti-IgM antibody or left untreated. The cells incubated with the recipient-derived plasma were also incubated with the anti-IgM antibody. After staining, the cells were washed in phosphate buffer and applied to FACScan flow cytometry (Becton Dickinson, Mountain View, CA, USA).

### 2.10. Statistical Analyses

The data were analysed with GraphPad Prism, Version 8.0 (GraphPad Software, San Diego, CA, USA). Paired or unpaired two-sided Student’s *t*-test and analysis of variance (ANOVA with Newman–Keuls post-test for multiple comparisons) were used where appropriate. Data are presented as mean values ± SEM (*n* is the number of samples, animals, or experiments). Significance was inferred at *p* < 0.05.

## 3. Results

### 3.1. Human Transcriptome Data

The published transcriptome data set GSE36059 (*n* = 411, indication biopsies) was used to analyse complement factor expression in human biopsy samples [27]. While complement upregulation was absent in nephrectomy and transplants without rejection (non-rejection (NR) samples), we found that the complement activation pathway (KEGG database pathway analysis) is significantly upregulated in TCMR, ABMR, and mixed rejection (Figure 1A). *C1QA*, *C1QB*, *C1S*, and *C2* (Figure 1B) were specifically upregulated in TCMR, whereas ‘downstream’ components C*4A*, *C5*, *C6*, and *C7*, as well as MBL-associated serine protease *(*M*ASP1* and MASP2), were unchanged (not shown). Since we hypothesised that apoptotic debris might be a trigger mechanism for complement expression, we next investigated the apoptosis gene pathway. Activation of apoptosis pathways was observed in all samples with rejection (TCMR, ABMR, mixed), but the highest expression was found in TCMR compared to non-rejection and nephrectomy (Figure 1C).

### 3.2. Animal Model

#### 3.2.1. General Functional Data and Histology

The data of the measurements performed on POD4 are given in in the supporting document (Appendix A). After TX, CrCl decreased and the occurring polyuria was paralleled by increased fractional Na^+^ and K^+^ excretion (aTX), whereas the decreased CrCl was not paralleled by polyuria or increased excretion of electrolytes in sTX. Impaired renal function was associated with histological features of AR, as published before [20,21]. In aTX, signs of TCMR such as interstitial and tubular infiltration, perivascular edema, and endothelialitis were seen at POD4. No such findings were visible in CTR and sTX (Appendix A).

#### 3.2.2. Local Gene Expression of Complement Factors

Microarray analysis showed that mRNAs related to cell-mediated immune response were upregulated in aTX compared with sTX on POD4 (19). However, the majority of genes associated with complement pathways were also upregulated (Figure 2). This indicates local synthesis of complement factors in the graft undergoing TCMR. Upregulated genes are listed in Appendix A and include *C1–C4* components. We used specific primer pairs and real-time PCR to validate the microarray data for the *C1q* alpha and beta chains (*C1qa, C1qb)*, *C2*, *C3*, and *C5*. Real-time PCR experiments showed upregulation of *C4bpa* as well as *C1qa* (8-fold), *C1qb* (19-fold), *C2* (65-fold), *C3* (30-fold), and *C4bpa* (110-fold) components with a significance level of *p* < 0.05 on mRNA level (Figure 3), whereas *C5* was not found to be upregulated in both data sets. The mRNA expression of MBL-associated serine protease (*Masp*) 1 showed a 1.9-fold induction (*p* < 0.05) while *Masp2* did not show significant changes in expression. With increasing intensity of rejection (time after transplantation [29]), the expression of complement components occurred or was higher on POD4 than on POD2 in aTX. Isografts without rejection did not show upregulated expression of *C1qa/b*, *C2*, *C5*, or *Masp1* and *Masp2* compared to healthy control kidneys (Figure 3). Significant upregulation was observed for complement 3 a receptor (*C3ar1*, fold change 1.9) and complement 5 a receptor (*C5ar1*, fold change 7).

#### 3.2.3. C4d Deposition Indicates Activation of the Complement System

In addition to mRNA data, grafts were analysed for deposition of *C4d* using a specific antibody. Controls and sTX showed *C4d* deposition in glomeruli, indicating independence from rejection (Figure 4A,B). In contrast, aTX showed additional positive linear *C4d* staining in peritubular capillaries (PTC) at luminal sites of some endothelial cells, as is usually the case with humoral rejection (Figure 4C) (14). Thus, in early TCMR, increased expression of complement components is associated with complement activation.

Usually, C4d deposition is noted secondary to alloantibody production and binding, which would indicate complement activation via the classical pathway. However, C3b staining was negative on POD4 (data not shown). Sensitisation to donor subjects or early production of alloantibodies 4 days after TX was excluded by analyses of the grafts and serum of recipient rats. On the one hand, we could not detect co-localisation of IgM and C4d deposition within the graft by immunofluorescence (Appendix A), and on the other hand, recipients’ sera were negative for circulating alloantibodies against donor cells (FACS analysis, Appendix A).

### 3.3. Structural Analysis and Apoptosis

As we excluded C4d activation via the classical pathway in TCMR, there should be an alternative trigger mechanism involved, such as activation by apoptotic debris. An upregulation of apoptosis-related pathways has been already observed in human TCMR transcriptome data (Figure 1C). Interestingly, we found monocytes/macrophages (Mphi) attached to C4d-positive sites in rat renal grafts with TCMR (Figure 4C,D). It has been shown that C1Q binds to apoptotic debris, thereby activating the complement system (30). Apoptosis is highly present in kidneys undergoing AR (31). Pathway analysis of rat data showed that the expression of several upregulated genes in TCMR was associated with allorecognition by CD8- and CD4-positive T cells and destruction of the allograft (Figure 5) [17]. Among the genes found to be upregulated in aTX, perforin, granzyme B, FAS, and tumour necrosis factor α were classified and visualised by Ingenuity Pathways Analysis with the function “apoptosis of cell lines” (Figure 5).

Further analysis showed that the expression of genes within the death receptor signalling pathway was significantly increased after aTX (Figure 6). Taken together, this indicates that apoptosis was induced in TCMR in humans as well as in rats.

To confirm an association of activation of the complement pathway and apoptotic cells, we performed ultra-structural analysis. While kidneys from controls and sTX showed normal structure (Figure 7A,B), apoptotic endothelial cells were frequently detected in TCMR. Moreover, we often observed phagocytosis of these apoptotic cells by macrophages (Mphi, Figure 7C,D).

## 4. Discussion

In this study, we show that TCMR is associated with the upregulation of genes involved in the complement pathway in kidney transplants. This was observed in humans as well as a rat renal transplant model (Figure 1 and Figure 2). Although the role of complement activated mainly via the classical pathway in ABMR is well-established and even a diagnostic and prognostic criterion, its role in TCMR is the subject of research. Analysis of published human transcriptome data indicated upregulation of complement genes not only in ABMR and mixed rejection, but also in TCMR, compared to nephrectomised kidneys and allografts without rejection (Figure 1). Since complement is involved in the removal of apoptotic cells, we hypothesised that complement activation in TCMR occurs in the context of the clearance of cell debris [30]. Because the complement pathway is highly conserved, we aimed to elucidate the role of renal complement activation in a defined rat model with TCMR in the absence of drug effects [31,32].

The upregulation of complement genes in kidneys with TCMR coincides with the activation of the complement system, as indicated by C4d staining. Controls and sTX showed C4 deposition only in glomeruli, indicating independency from rejection (Figure 4). This is in agreement with Collins et al. [33]. Moreover, relevant complement deposition associated with ischemic/reperfusion injury (IRI) was excluded by including an sTX group. However, it has been postulated that IRI in heart, lung, intestine, and muscle can activate the complement system via the classical pathway [34]. Recent work suggests the relevance of naturally occurring IgM self-reactive antibodies (nAb) in the induction of complement activation and progression of IRI [12]. The authors discovered in lung transplantation by an elegant approach using application of C2SCFV-CRRY, which on the one hand prevents nAb binding to C2 and on the other hand impedes C3 activation by direct inhibition, that IRI can be significantly reduced. In line with our results, they observed increased C2 expression after allogeneic lung transplantation and were able to ameliorate IRI and rejection by application of *C2SCFV-CRRY*. After cardiac transplantation, it has been observed in post-ischemic hearts that, for example, expression of annexin IV and C2 increased, and its recognition by nAb is associated with graft IRI [13]. For renal IRI, complement activation via the alternative pathway independent of C4 has been postulated [35]. However, using an antibody directed against C3d, we were not able to detect C3d deposition outside the glomeruli on POD4 after sTX, indicating a lower and different IRI-dependent complement activity in this model than in TCMR. Others found significantly increased C3 expression within 5 h after rat renal sTX [36]. However, we cannot exclude that IRI during sTX was not strong enough in our model to trigger relevant C2 expression or that complement expression has already been normalised at POD4 [37].

Gene expression data of complement factors indicate a local (i.e., intrarenal) synthesis of these factors. The ability of renal cells, such as mesangial or tubular cells, to express complement components in response to rejection has been reviewed by Sacks et al. [38,39]. Interestingly, in addition to C6, mRNA expression of the other components involved in MAC formation, such as C5 and C7–9 complement components, was not increased in TCMR. Consistent with these results, human transcriptomic data showed comparable expression differences in nephrectomy, graft biopsies without rejection, and biopsies with T cell-mediated rejection (Figure 1). The only observed difference from the rat data was *C4bpa* expression, which was significantly upregulated in rat kidneys with TCMR but not in the human samples. The upregulation of counter-regulatory *C4bpa* could be related to the fact that TCMR in rats was more intense and not attenuated by immunosuppressive medication (maintenance therapy) given in humans. This is matched by the fact that *C4bpa* expression is significantly stronger at POD4 than at POD2, so it could correlate with the intensity of rejection [29].

While we were able to detect C4d deposition in grafts undergoing TCMR, we were unable to detect alloantibodies either circulating or deposited in the graft. Three different pathways of complement activation lead to C4d deposition. C1q initiates activation via the classical pathway by binding to the Fc portion of an antigen-bound antibody, which triggers the subsequent formation of the membrane attack complex [40]. C1q has additional immune functions since professional phagocytic cells such as Mphi have the C1q receptor (C1qR) [41]. Binding of C1q to endothelial cells upregulates adhesion molecules and stimulates the production of chemoattractants, resulting in migration of leukocytes [40]. The involvement of C1q can be pro- and anti-inflammatory. For example, the classical complement pathway is needed for the prevention of inflammation and for clearing inflammatory mediators [42]. Notably, autoimmunity in C1q^−/−^ mice is characterised by glomerulonephritis and the distinct presence of apoptotic bodies and IgG deposits [43]. As C1q binds to apoptotic bodies and is recognised by Mphi, Mphi isolated from C1q^−/−^ mice have a reduced capacity to phagocytise apoptotic cells [44,45]. In addition, *C1q* deficiency, but not C3 deficiency, has been shown in mice to be associated with systemic lupus erythematosus—a disease caused by defective clearance of dead and dying cells [46]. Interestingly, we measured increased *C1q* mRNA-level in aTX only—the same group that exhibited increased numbers of apoptotic cells. We also detected apoptotic cells that were phagocytised by Mphi (Figure 7).

The lectin pathway uses an opsonin, mannose-binding lectin (MBL), and ficolins instead of C1q. This pathway is activated by mannose residues on the surface of the pathogen. These activate the MBL-associated serine proteases, MASP-1 and MASP-2, which can cleave C4 into C4a and C4*b* and cleave C2 into C2a and C2b. Notably, monoclonal anti-*C1qR* inhibits the MBL-dependent stimulation of phagocytic activity in Mphi, indicating that the assessed activity is mediated by C1qR [41]. Interestingly, there must be additional signalling, e.g., related to the cellular surface, that determines whether C1q leads to superoxide production in neutrophils or stimulates phagocytosis in Mphi [41]. However, *Masp-1* and *Masp-2* mRNA did not increase in TCMR in either humans or rats (Figure 1 and Figure 3).

Tubulitis and endothelialitis are characteristic features of TCMR—features that were associated with impaired renal function (e.g., polyuria, increased FE_Na_+) and were observed in aTX only (SDC-1). Activated, endothelium-adherent cytotoxic T cells damage PTCs by lysis or the induction of apoptosis of endothelial cells. In contrast, typical pathological features of ABMR include the swelling of endothelial cells, rupture and denudation of membranes, and intra- and extra-capillary fibrin deposition [47]. Lipták et al. described the parallel existence of lysis and apoptosis in PTCs from recipients with ABMR. Polymorphonuclears and/or mononuclear cells in PTCs, presumably recruited by chemotactic complement fragments or chemokines, are putative (histologic) features of ABMR [48]. Lipták et al. discovered that Mphi were the primary immune cell type present in PTCs of ABMR-affected grafts [47]. They suggested that their main function is to scavenge luminal debris.

Our data indicate that C4d deposition in our model is a protective strategy of the kidney for the clearance of apoptotic cells. Since we did not observe deposition of C3d at the sites of C4d, the activation of the complete complement cascade does not appear to be necessary for this process. In this case, regulatory factors controlling complement activation should be involved. Complement inhibitor component 4-binding protein (C4BP), a soluble and circulating protein, inhibits the classical and lectin pathway of complement activation [49]. It inhibits the formation of C3 convertase and accelerates its decay, which may explain the absence of C3d staining. *C4bp* alpha chain mRNA is one of the most upregulated genes in rejection. It has also been shown that C4BP can bind to apoptotic cells together with protein S [50]. Therefore, one can speculate that C4BP is the responsible factor that inhibits further activation of the complement cascade and membrane attack complex formation in our model.

## 5. Conclusions

In summary, we show that upregulation of genes associated with the complement pathway shortly after transplantation in kidney grafts undergoing TCMR coincided with apoptosis and the activation of the complement system. Comparable upregulation of complement and apoptosis genes was found in transcriptomic data from patients with TCMR. We hypothesise that apoptotic cells are the trigger mechanism for complement activation in TCMR and propose a role for complement as a recognition marker for Mphi to clear the graft of apoptotic debris.

## Figures and Tables

**Figure 1 cells-10-03499-f001:**
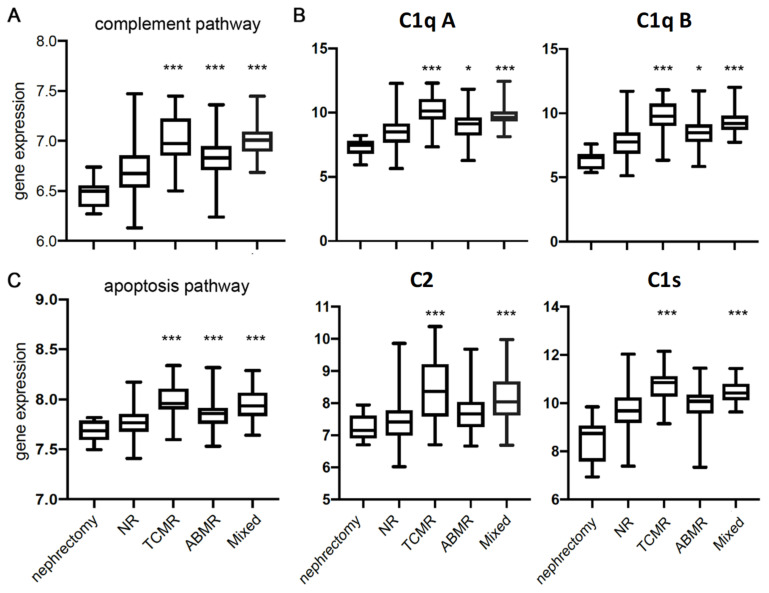
Transcriptome analysis of human kidney transplant biopsies. GSE36059 with *n* = 411 indication biopsies. (**A**,**C**) KEGG pathways. Gene expression shows pathway activity (mean of sum of all genes). (**B**) Individual genes of the complement pathway. NR = non-rejection. * indicates *p* < 0.05 compared to nephrectomy, *** indicates *p* < 0.001 compared to nephrectomy.

**Figure 2 cells-10-03499-f002:**
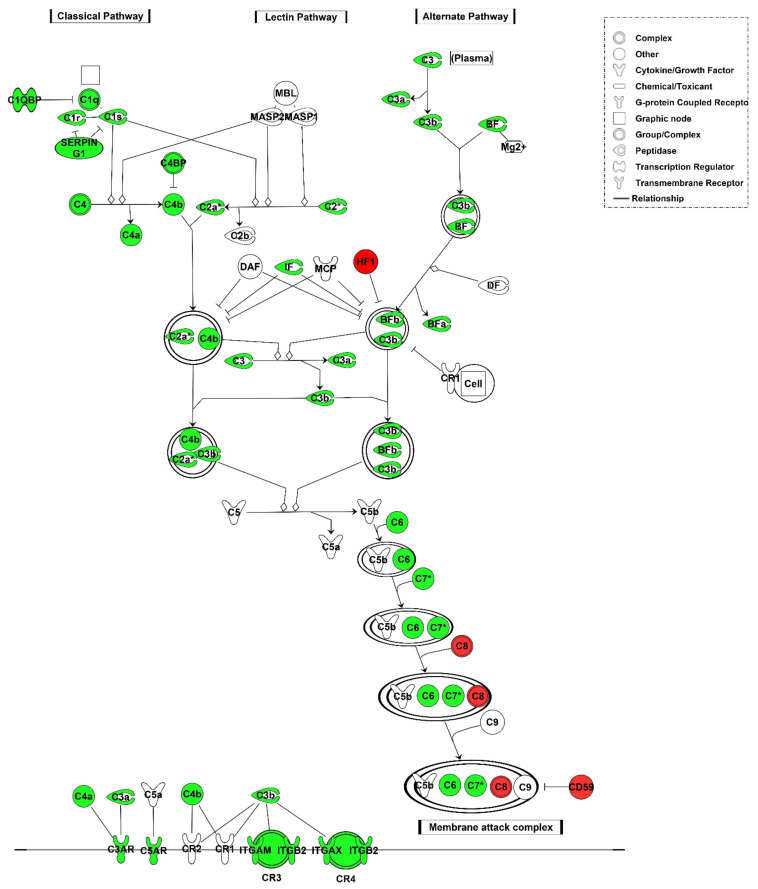
Complement pathway. Ingenuity Pathways Knowledge Base was used to display pathways containing multiple differently expressed genes. Significantly regulated genes after allogeneic transplantation with TCMR compared to syngeneic transplantation without rejection were mapped with the complement pathway. Upregulated genes are highlighted in green, downregulated genes are highlighted in red.

**Figure 3 cells-10-03499-f003:**
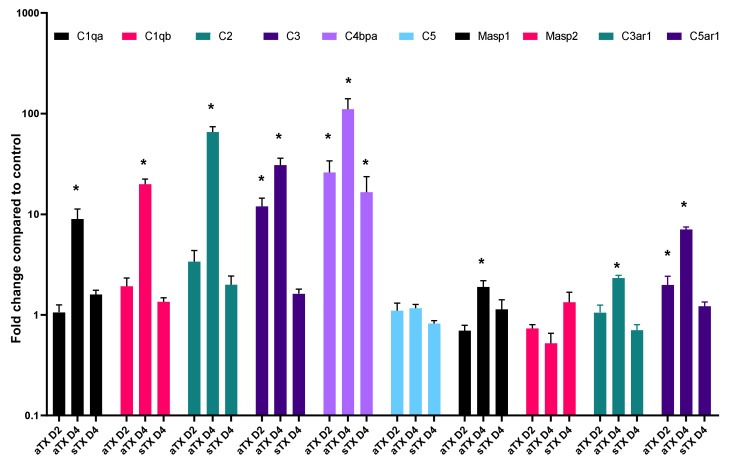
Allogeneic transplantation induces expression of complement components. The mRNA expression of selected members of the complement pathway was analysed in kidneys on day 2 after aTX (aTX D2), day 4 after aTX (aTX D4), or on day 4 after sTX (sTX D4) by real-time PCR. The expressions relative to the control kidney were calculated using the 2^−ΔΔCt^ method (*n* = 3–5/group). One-way ANOVA was used to calculate significant differences in expression level compared to control kidney (*p* < 0.05) and is marked with *.

**Figure 4 cells-10-03499-f004:**
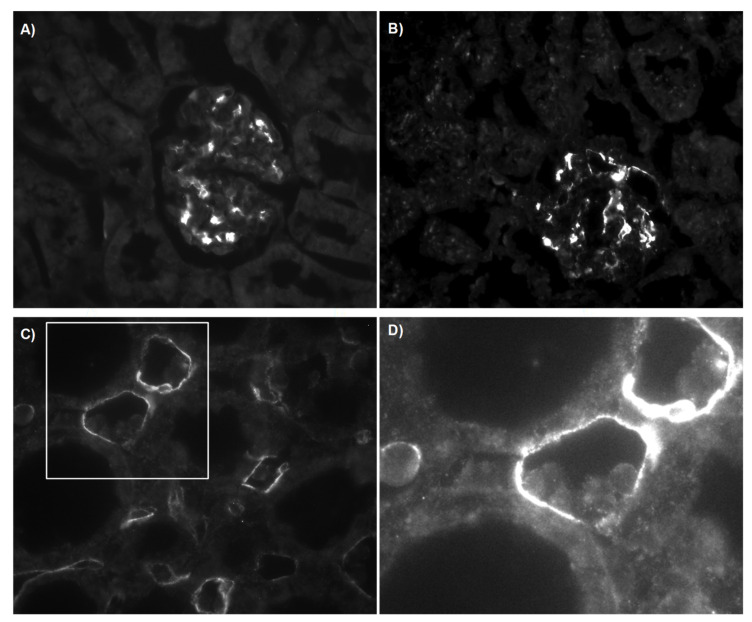
Representative immunohistochemical staining for C4d. Cryosections from CTR (**A**), sTX (**B**), and aTX (**C**) kidneys were stained using an antibody directed against C4d. After incubation with an Alexa488 labelled secondary antibody, signals were visualised by an epifluorescence microscope. Positive C4d staining was observed in the glomeruli of CTR and sTX kidneys (**A**,**B**). Additional staining was only observed after allogeneic transplantation (**C**). Magnification of the marked area from C shows that cells were attached at C4d-positive sites (**D**).

**Figure 5 cells-10-03499-f005:**
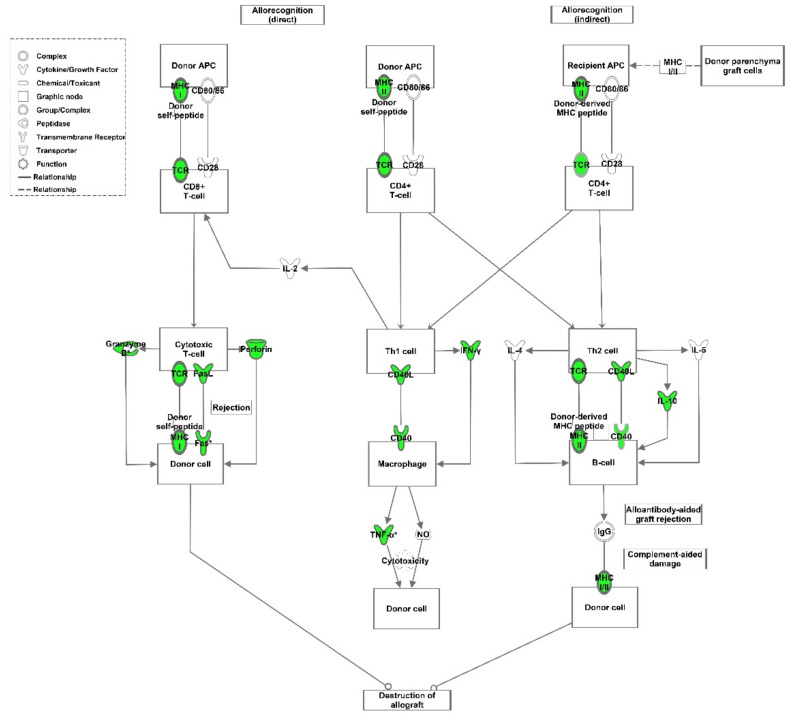
Induced expression of genes involved in allorecognition signalling. Ingenuity Pathways Knowledge Base was used to display pathways containing multiple differently expressed genes. Significantly regulated genes after allogeneic transplantation compared to syngeneic transplantation were mapped with the allograft signalling pathway. The lines indicate the type of relationships. Solid lines indicate a direct and dotted lines an indirect interaction. Upregulated genes are highlighted in green.

**Figure 6 cells-10-03499-f006:**
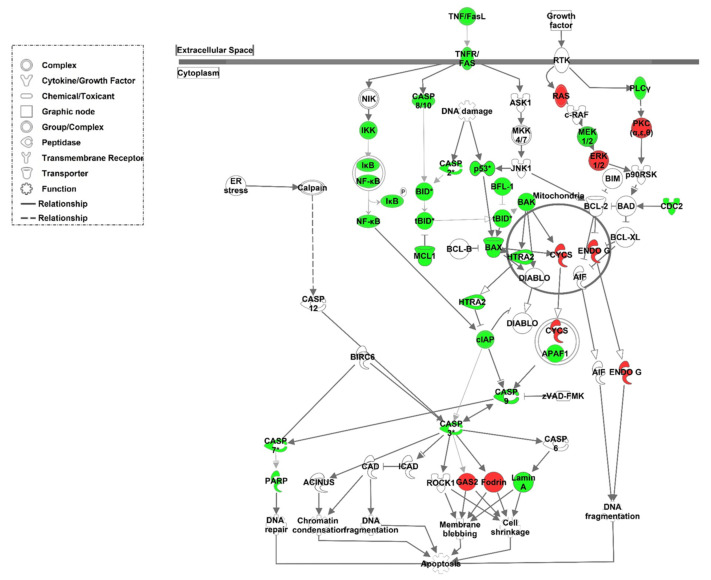
Induced expression of genes involved in death receptor signalling. Ingenuity Pathways Knowledge Base was used to display pathways containing multiple differently expressed genes. Significantly regulated genes after allogeneic transplantation compared to syngeneic transplantation were mapped with the death receptor signalling pathway. The lines indicate the type of relationships. Solid lines indicate a direct and dotted lines an indirect interaction. Upregulated genes are highlighted in green, downregulated genes are highlighted in red.

**Figure 7 cells-10-03499-f007:**
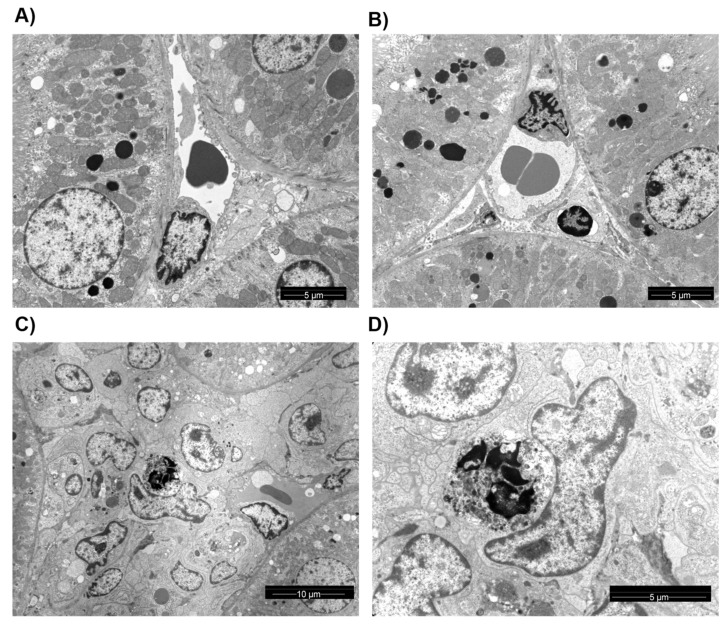
Ultra-structural analysis. Kidney section from CTR (**A**), sTX (**B**), and aTX (**C**,**D**) transplanted kidneys were used for ultrastructural analysis. The control and syngeneic transplanted kidneys showed normal morphology. Allogeneic transplanted kidneys with TCMR showed apoptosis and activation of Mphi (marked by a white arrow) engulfing apoptotic debris (marked by the black arrow).

## Data Availability

The data is deposited at GEO, GEO Series accession number GSE6497.

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
