# Peer review of "C4d Deposition after Allogeneic Renal Transplantation in Rats Is Involved in Initial Apoptotic Cell Clearance"

_cells, 2021, doi:10.3390/cells10123499_

Round 1
Reviewer 1 Report
The authors have made serious efforts to address the comments raised by the reviewers. Minor changes in spelling and figure presentation is needed (for example reduce fort size in Figures 4 and 6). Figure 5 is still hard to follow. Consider breaking in two figures.
Author Response
We thank the reviewer for the comments and suggestions. We have reduced the font size in Figure 4 and 6 (now figure 7).
We have also revised figure 5 and now it is splitted in figure 5 and figure 6.
Reviewer 2 Report
Concerns were adequately addressed by the authors
Author Response
We thank the reviewer for this comment.
This manuscript is a resubmission of an earlier submission. The following is a list of the peer review reports and author responses from that submission.
Round 1
Reviewer 1 Report
The main observation in this manuscript is the fact that “pure” T cell-mediated acute kidney allograft rejection (AR) or as it is called in human T cell-mediated rejection (TCMR) is with the complement activation. The authors suggest and provide evidence that such AR complement activation is related to an apoptosis process and correlate with the involvement of apoptotic pathways.
Main concerns:
- the organization of this manuscript can be improved as some results in rats are shown first and then followed by human results and then back to rat results. The problem is that human results showed TCMR group, antibody-mediated rejection (ABMR) group, and then mixed TCMR/ABMR group. Consequently, it would be better to show and explain human results first and clearly show the human results with TCMR group. The authors may show the lack of donor-specific antibodies in TCMR group of presented patients with simultaneous evidence for the complement activation. This arrangement would provide a logical introduction for the experiments in rats focused on “pure” T cell-mediated AR and complement activation. The authors may show complement activation in TCMR group, ABMR group and mixed TCMR/ABMR group, but then the focus should be on the TCMR group.
- The authors show that the activation of complement on day 4 after kidney transplantation occurs because of the classical cellular allograft rejection (AR) driven by T cells. It is true and correct (also new) that the activation of complement and the deposition of C4d on day 4 is related to an apoptosis process in the developing AR. The eventual weakness of this paper is the fact that they present results only on days 2 and 4. Indeed, it would be more interesting to present the kinetics on more days after kidney allograft transplantation. It would be better to show the kinetic of apoptosis in kidney allografts vs. syngeneic kidney transplants on days 2, 4, 6 and 8 after transplantation. Such kinetics would show the difference in apoptotic cells in kidney transplants between an allograft and syngeneic graft. Increasing apoptosis of allogeneic cells damaged by host T cells would correlate with the activation of the complement components.
- The Materials and Methods section should have more explained about the kidney transplantation method. In an Animal model section there is not even kidney transplantation expression.
- The sentence on page 6 (top page) should be “As published before, in aTX signs of AR occur on POD4 as marked by interstitial and tubular infiltrations, perivascular edema, and endothelialitis”
- The first paragraph of the Discussion section should be focused on the importance of findings in the current manuscript: correlation of pure T cell-mediated rejection with complement activation. The main question remains whether the complement activation is related to the T cell-mediated immune response or rather detrimental changes made by T cell-mediated damage. The authors after stating about AR and complement activation rather talk about other models. What is the connection between pure T cell response and complement activation? The first paragraph does not explain the contribution of the current work to the new discovery. After reading this section, the reader still does not know what the connection between T cell-mediated rejection is and complement activation.
- The last paragraph of the Discussion section should be developed and actually to be presented as the first paragraph in the Discussion section. Similarly, the Conclusion writes “we show that up-regulation of genes associated with the complement pathway shortly after TX in kidney grafts undergoing AR coincided with apoptosis and the activation of the complement system”. In fact, the last sentence in the Conclusion “ We hypothesize that apoptotic cells are the trigger mechanism for complement activation in AR and propose a role for complement as a recognition marker for Mphi to clear the graft of apoptotic debris” should be used as the main subject of this manuscript and the main subject of evidence provided and the main subject for the Discussion. The overall Discussion section should be reorganized to address the hypothesis that “T cell-mediated acute rejection triggers apoptosis of kidney allogeneic cells which initiate complement activation contributing to the apoptotic process.” This is why it would be helpful to show the kinetics of apoptosis in kidney allografts vs. syngeneic kidney transplants on days 2, 4, 6 and 8 after transplantation. Such kinetics would show the difference in apoptotic cells in kidney transplants between an allograft and syngeneic graft.
Additional Comments:
- Authors should ensure complement abbreviations are following the accepted abbreviations established by the International Complement Society, see PMID:
- Introduction could benefit from short description of complement-mediated diseases affecting the kidneys (e.g., Membranoproliferative glomerulonephritis, Atypical Hemolytic Uremic Syndrome) to emphasize the unique susceptibility of the kidneys to complement-mediated attack.
- Introduction could also benefit from a brief comparison between acute antibody-mediated rejection and acute cellular rejection.
- For Figure 1, please clarify what the red, teardrop shape indicates. Though this symbol is described in Figure 5, it should also be described in Figure 1.
- Figure 1: Some of the labels overlap each other. Correction required to make it legible.
- Authors should define sTX and aTX within manuscript text. As of now, these abbreviations are defined in the supplemental material only.
- Label y-axis and include statistical analysis for Figure 2.
- Figure 2: Label the y-axis. Add relevant statistical analysis.
- When describing results in the text, p-values should be included for each reported finding.
- The following statement is not supporting by Figure 3 because no data for C8 is given in Figure 3: “…whereas the expression of “downstream” components such as C5 or C8 was comparable to the expression levels of transplanted patients without rejection or with nephrectomy (Figure 3).”
- It is not clear what the statistical analysis in Figure 3 describe. It appears from reading the text, that nephrectomy and NR are being compared to all other groups, but this is not clear in Figure 3. Because of this lack of clarity, it is difficult to assess the validity of the results regarding these results.
- Please define NR in Figure 3. Please describe what the descriptive statistics represented by the box and whisker plots indicate in the Figure legend.
- Figure 4 may benefit from a representative H&E stain to better appreciate renal structures where C4d is present.
- It is stated in the text that C3d staining was not observed, but it would be a robust comparison to show this data so readers can compare it to C4d deposition (as shown in Figure 4).
- Correct to refer to Figure 4 on line 241.
- Figure 5 could benefit from labeling the structures, including evidence of apoptosis.
- Line 292-293 is written awkwardly and may be missing a word.
- Regarding the statement made on lines 301-303, authors should describe why rats were not given immunosuppressive drugs to simulate human response. Authors should also comment on the differences between rat and human complement systems.
- Be consistent with capitalization of “C4BPA” throughout.
- Lines 325-326, authors mention alternative pathway, but this does not add any value to the conclusions. Authors should elaborate on the involvement of the alternative pathway in renal pathology or remove this statement.
Reviewer 2 Report
The study by Reuter and colleagues has used an animal model as well as transcriptomic analysis from human samples in order to dissect the role of complement cascade in allograft rejection. The manuscript is incomprehensible and hard to follow. Major comments: 1) The authors need to clearly describe results in each section including the abstract and specify whether results are derived from the rat model or whether they are derived from human samples. 2) The authors should provide a brief description of the Microarray methodology at least mentioning the tissue they used from animals. Simple referencing a previous study is not helpful. 3) The authors did not detect C3d deposition and provide an explanation by the presence of C4BP but they should at least attempt to detect or quantify it in the rat model. 4) Did authors measure or detect C3AR1 or C5AR1 both of which have been shown to be involved in allograft survival? 5) Did the authors investigate the expression of other rat complement control proteins CD55, and Crry? They only show results for CD59. They should consider adding immunohistochemistry for complement control proteins in renal tissue sections. 6) Figure 5 is unreadable. The authors should either break down in two figures of simplify. 7) Figures 1 and 5. The authors should change the colour of upregulated and down regulated genes (green is normally used for upregulation and red for downregulation). 8) Figure 4. The authors should include a Ig only control and quantify fluorescence intensity. Minor comments: 1)Re-write abstract clearly describing results. 2) Carefully proof read for minor typos. 3) line 179: what do the authors mean by ‘canonical complement pathway activation? There are mainly three pathways (classical, alternative and lectin) all of which are canonical.